# Aging-Affected MSC Functions and Severity of Periodontal Tissue Destruction in a Ligature-Induced Mouse Periodontitis Model

**DOI:** 10.3390/ijms21218103

**Published:** 2020-10-30

**Authors:** Kyaw Thu Aung, Kentaro Akiyama, Masayoshi Kunitomo, Aung Ye Mun, Ikue Tosa, Ha Thi Thu Nguyen, Jiewen Zhang, Teisaku Kohno, Mitsuaki Ono, Emilio Satoshi Hara, Takuo Kuboki

**Affiliations:** 1Department of Oral Rehabilitation and Regenerative Medicine, Okayama University Graduate School of Medicine, Dentistry and Pharmaceutical Sciences, Okayama 700-8558, Japan; pidh4nf2@s.okayama-u.ac.jp (K.T.A.); de18022@s.okayama-u.ac.jp (M.K.); aungyemun@s.okayama-u.ac.jp (A.Y.M.); de421035@s.okayama-u.ac.jp (I.T.); thuharhm@gmail.com (H.T.T.N.); zhang-2019@s.okayama-u.ac.jp (J.Z.); p8bb7mk2@s.okayama-u.ac.jp (T.K.); kuboki@md.okayama-u.ac.jp (T.K.); 2Department of Molecular Biology and Biochemistry, Okayama University Graduate School of Medicine, Dentistry and Pharmaceutical Sciences, Okayama 700-8558, Japan; mitsuaki@md.okayama-u.ac.jp; 3Department of Biomaterials, Okayama University Graduate School of Medicine, Dentistry and Pharmaceutical Sciences, Okayama 700-8558, Japan; gmd421209@s.okayama-u.ac.jp

**Keywords:** mesenchymal stem cell, aging, tissue destruction, periodontitis, immunomodulation, bone resorption

## Abstract

Mesenchymal stem cells (MSCs) are known to play important roles in the repair of lost or damaged tissues and immunotolerance. On the other hand, aging is known to impair MSC function. However, little is currently known about how aged MSCs affect the host response to the local inflammatory condition and tissue deterioration in periodontitis, which is a progressive destructive disease of the periodontal tissue potentially leading to multiple tooth loss. In this study, we examined the relationship between aging-induced impairment of MSC function and the severity of periodontal tissue destruction associated with the decrease in host immunomodulatory response using a ligature-induced periodontitis model in young and aged mice. The results of micro computerized tomography (micro-CT) and histological analysis revealed a more severe bone loss associated with increased osteoclast activity in aged (50-week-old) mice compared to young (5-week-old) mice. Immunostaining analysis revealed that, in aged mice, the accumulation of inflammatory T and B cells was higher, whereas the percentage of platelet-derived growth factor receptor α (PDGFRα)^+^ MSCs, which are known to modulate the apoptosis of T cells, was significantly lower than in young mice. In vitro analysis of MSC function showed that the expression of surface antigen markers for MSCs (Sca-1, CD90, CD146), colony formation, migration, and osteogenic differentiation of aged MSCs were significantly declined compared to those of young MSCs. Moreover, a significantly higher proportion of aged MSCs were positive for the senescence-associated β galactosidase activity. Importantly, aged MSCs presented a decreased expression of FAS-L, which was associated with a lower immunomodulatory property of aged MSCs to induce T cell apoptosis in co-cultures compared with young MSCs. In summary, this is the first study showing that aging-induced impairment of MSC function, including immunomodulatory response, is potentially correlated with progressive periodontal tissue deterioration.

## 1. Introduction

Aging is an inherent physiological process of life, characterized by impairment of biological processes in living organisms with loss of function and multiple diseases, and has been recently introduced as a new disease in the International Classification of Diseases released by the World Health Organization [1]. From the biological perspective, aging is associated with changes in tissues and cells, resulting, for instance, in the accumulation of various molecules and cellular damage over time, and a consequent impaired homeostasis and function and reduced capacity to respond appropriately to injury [2,3]. 

Recently, the use of stem cells has gained wide popularity as a potential therapeutic for tissue regeneration and management of aging-related disorders and diseases [4,5]. Mesenchymal stem cells (MSCs) are one of the multipotent stromal cell sources for tissue regeneration. They can be isolated not only from the bone marrow, but also from several adult tissues, including adipose tissue, and those in the orofacial region, such as dental pulp, gingiva, and periodontal ligament [6,7,8]. One of the most important abilities of MSCs in tissue regeneration is the capability for multiple differentiation, such as into osteoblasts, adipocytes, chondrocytes, and myoblasts. Local immune tolerance has also been regarded as another fundamental characteristic of MSCs in tissue regeneration [9]. Two different mechanisms of immune response modulation by MSCs have been clarified. The first is related to indirect inhibition of the ability of inflammatory cells (i.e., T lymphocytes, dendritic cells, natural killer (NK) cells, and B lymphocytes) to proliferate and secrete inflammatory cytokines, through the release of anti-inflammatory cytokines, including transforming growth factor-beta (TGF-β), hepatocytes growth factor (HGF), prostaglandin E2 (PGE2), indoleamine 2,3-dioxygenase (IDO), nitric oxide (NO), or human leukocyte antigen-G5 (HLA-G5) by the MSCs [10,11,12,13,14]. The other inhibition mechanism is based on a direct cell-to-cell contact that induces immune cell apoptosis through the FAS/FASL, leading to the regulatory T cell differentiation and immune tolerance [15].

During physiological tissue repair, cytokines, chemokines, and growth factors such as the stem cell factor (SCF), granulocyte colony stimulation factor (G-CSF), and stromal cell-derived factor 1 (SDF-1, CXCL12) are released at the injured tissue and recruit circulating MSCs to the inflammatory site via two distinct molecular pathways, namely, the SDF-1/CXCR4 and MCP-1/CCR2 axis [8,16]. Therefore, due to the multiple differentiation capability, immunomodulatory property, and endogenous cell recruitment ability, MSCs have been reported to have crucial roles in tissue regeneration [17,18], management of immune-mediated diseases [19,20,21], and cell-based bone augmentation [22]. 

A number of studies have reported that aging affects stem cells by impairing their protective and regenerative capacities, by inducing cellular senescence and decreasing stem cell numbers and functions [23,24,25,26,27]. Several in vivo studies in various species (rodents, monkeys, and humans) have confirmed the decreases in the number of cells harvestable from the bone marrow and in the expression of MSC antigen surface markers (e.g., Sca-1, CD90, CD146), as well as in diverse MSC functions, including cell proliferation and differentiation, colony formation (CFU-f), and immunomodulation [28,29]. 

Periodontitis is an inflammatory disease of tooth-supporting tissues caused by specific microorganisms resulting in progressive destruction of the periodontal ligament and alveolar bone with the periodontal pocket formation and/or gingival recession, which eventually can lead to multiple tooth loss and a consequent decrease in oral function and quality of life [30]. The pathophysiology of periodontitis has been regarded as the host response to bacteria with a local inflammatory reaction that stimulates the innate immune system. Prolonged inflammation exaggerates the release of a broad spectrum of proinflammatory cytokines and mediators, such as interleukin-1 (IL-1) and tumor necrosis factor-alpha (TNF-α), which induce osteoclastogenesis via the RANKL–RANK–OPG pathway, and in turn, can lead to connective tissue and alveolar bone destruction [31,32,33]. Epidemiological evidence showed that the prevalence of periodontal disease might range from 20% to 50% worldwide [34], and in the United States, the distribution of mild, moderate, and severe periodontitis in 64.7 million adults is known to be 8.7%, 30.0%, and 8.5%, respectively [35]. Notably, the occurrence and severity of periodontal tissue destruction are known to increase with age [35,36,37].

Herein, we hypothesized that the decrease in host response and capability of periodontal tissue repair in aged individuals could be associated with aging-induced phenotypic changes in MSCs. Therefore, the objective of this study was to investigate the correlation between the severity of periodontal bone destruction and aging-affected functions, including the immunomodulatory properties, of MSCs by in a ligature-induced periodontitis model in young and aged mice.

## 2. Results

### 2.1. Severe Bone Resorption after Ligation in Aged Mice

A periodontitis model using ligatures was developed in young (5-week-old) and aged (50-week-old) mice, according to the experimental design shown in Figure 1A,B. The micro computerized tomography (micro-CT) analysis showed more severe bone resorption at both day-3 and day-10 post-ligation in the aged compared to young mice (arrowhead, Figure 1C). Quantitative analysis of bone resorption area and depth in the lingual side confirmed the significantly higher amount of lost bone area in the aged compared to young mice at day-3 and day-10 after ligation (Figure 1E, left graph). There was also a significantly higher bone loss in depth (mesial, distal) in the aged (1.037 ± 0.095 mm, 0.643 ± 0.008 mm) compared to young mice (0.558 ± 0.034 mm, 0.374 ± 0.060 mm) respectively, at day-10 after ligation (Figure 1E, middle and right graphs). Of note, even though the control groups with no ligature in young and aged mice showed a significant difference at baseline, a two-way ANOVA analysis revealed that there was a significant association between age and ligation-induced periodontitis.

Histological analysis with hematoxylin and eosin (HE) stained sections confirmed the increase in periodontal ligament space in the aged mice (35.28 ± 6.39) vs. young mice (27.37 ± 2.14) at day-10 after ligation (Figure 2A, bidirectional arrows) and the formation of small pits indicating the activity of osteoclasts (Figure 2A, arrowheads). Furthermore, aged mice showed less dense alveolar bone compared to the young ones (Figure 2A). In fact, the number of tartrate-resistant acid phosphatase (TRAP)^+^ osteoclasts was significantly higher in the aged (76.91 ± 7.75) than in the young mice (59.96 ± 8.39) at day-10 after ligation (Figure 2B).

### 2.2. Increased Inflammatory Cell Accumulation at the Periodontal Bone Destruction Area in Aged Mice

Analysis of the presence of T and B cells in the ligature-induced periodontitis site confirmed the increased number of CD3^+^ pan T cells around the furcation area of the first molar region both at day-3 and day-10 in young and aged mice. Notably, aged mice showed a remarkably higher number of CD3^+^ T cells compared to young ones at day-3 and day-10 after ligation (Figure 3A). Similarly, the number of B220^+^ B cells (day-3, day-10) were significantly higher in aged (8.02 ± 0.65, 10.92 ± 2.38) than in young mice (3.99 ± 0.65, 5 ± 1.24), respectively (Figure 3B). Furthermore, after ligation, the levels of inflammatory cytokines, including *Il-1β*, *Tnf-α*, *Ifn-γ*, were all significantly increased at day-10 in aged mice (Appendix A). 

### 2.3. Decreased MSCs Number in Aged Mice

MSCs are known to participate actively in host immunomodulation by directly interacting with T and B cells [38,39]. Therefore, immunohistochemical analysis was performed to evaluate the distribution of platelet-derived growth factor receptor α (PDGFR)α^+^ MSCs in the periodontitis-affected milieu. In young mice, the number of PDGFRα^+^ cells was increased after ligation at day-10, while in aged mice, there was no significant variation. Moreover, aged mice presented fewer PDGFRα^+^ cells (day-3: 4.05 ± 0.48, day-10: 4.32 ± 0.87) compared to young mice (day-3: 4.67 ± 0.81, day-10: 7.86 ± 1.62) after ligation (Figure 4).These results suggest that the decreased number of PDGFRα^+^ MSCs could be one of the factors determining the increase in the number of inflammatory T and B cells, and the consequent increase in osteoclast-directed bone resorption in the ligature-induced periodontitis in aged mice.

### 2.4. Functional Impairment of Aged MSCs

The decreased number of MSCs at the injured site in aged mice could be associated with an impaired migration ability of aged MSCs. In addition, other MSC functions, including immunomodulation, could also be affected by aging. Therefore, we analyzed the expression pattern of stem cell markers and the major functions of MSCs, including colony formation (CFU) ability, migration, differentiation, and immunomodulation. 

As shown in Figure 5A, MSCs from aged mice showed a significant decrease in the CFU-f colony-forming ability compared to young MSCs. Moreover, a scratch assay revealed that the migration speed of young MSCs was significantly faster than that of aged MSCs at 48 h after scratching (Figure 5B). Analysis of the cell surface antigen expression by flow cytometry showed that both MSCs derived from young and aged mice were highly positive for the MSC-associated markers, Sca-1, CD 90, CD146, but not for hematopoietic stem cell markers, CD14 and CD34. Notably, however, the percentage of MSC surface markers, Sca-1, CD 90, and CD 146, was significantly lower in aged compared to young MSCs (Figure 5C). On the other hand, aged MSCs showed a significantly higher activity of cell senescence-associated β-galactosidase (SA-β gal) compared to young MSCs (Figure 5D). Additionally, aged MSCs presented a markedly lower osteogenic differentiation ability as demonstrated by alizarin red-S staining for mineral deposition, as well as by the analysis of gene expression levels of osteoblast markers, Alkaline phosphatase (*Alp*) and Osteocalcin (*Ocn*) (Figure 6A). Conversely, oil red-O staining for lipid droplets and the gene expression levels of the adipogenic markers, Peroxisome proliferator-activated receptor gamma (*Pparγ*) and Lipoprotein lipase (*Lpl*), were significantly increased in aged MSCs (Figure 6B). These results indicate that aging strongly downregulates the osteogenic differentiation capacity of MSCs, and could, therefore, in part explain the decreased ability of aged mice to repair the periodontitis-associated bone loss. In addition, trabecular bone volume in the femur was significantly decreased in aged mice (Appendix A).

To evaluate the immunomodulatory property of young and aged MSCs, FAS-L expression was analyzed by flow cytometric analysis. As shown in Figure 6C, the expression of FAS-L was reduced in aged MSCs compared to young MSCs. In addition, the percentage of Annexin V^+^ apoptotic CD4^+^ T cells was significantly higher when the cells were co-cultured with either young or aged MSCs (Figure 6C). Note that the percentage of Annexin V^+^ apoptotic CD4^+^ T cells was significantly lower in the co-culture with aged MSCs, compared to that in the co-culture with young MSCs, indicating a lower immunomodulatory function of aged MSCs. These results could in part explain the higher number of inflammatory T and B cells observed at the periodontitis site. 

## 3. Discussion

Periodontitis is defined as an inflammatory disease of the tooth-supporting tissues caused by specific microorganisms, resulting in progressive destruction of the periodontal ligament and alveolar bone with the periodontal pocket formation and/or gingival recession [30]. Available epidemiological evidence shows that the occurrence and severity of periodontal destruction increase with age [37]. The etiology of periodontitis has been well understood by the imbalance between the host immune system and the pathogens. Thus, the removal of the pathogens is the initial and fundamental treatment strategy. However, the involvement of the host immune system and its regenerative capacity have not been fully elucidated. Here, we developed the ligature-induced periodontitis model in young (5-week-old) and aged (50-week-old) mice in an attempt to correlate the aging-affected severity of periodontal tissue destruction, the degree of inflammation at the periodontitis environment, and the function of MSCs. The aged mice used in this study would represent humans in their third or fourth decades of life, during which the prevalence of severe periodontitis is known to be high but further remain relatively stable at older ages [40]. We examined the severity of bone resorption in mice at different ages, including 5, 20, 35, 50, 65, and 80-week-old mice, by micro-CT analysis. Interestingly, there was no significant difference in bone resorption in mice over 50 weeks of age (Appendix A). Therefore, the 50-week-old mice were selected to investigate the pathophysiology of periodontitis relative to middle-aged patients. Notably, in the unligated control groups, physiological bone resorption was found to be higher in the aged than in the young mice, which is consistent with previous reports [41] and strongly supports the consistency of the methods. 

Histologically, loss of attachment and deep pocket formation are the hallmarks of periodontitis [42]. Both chronic periodontitis and aggressive periodontitis show the clinical features of bone loss and attachment loss in response to colonization of the tooth surface by a bacterial biofilm [30]. Bone resorption is a consequence of periodontitis due to the infiltration of inflammatory cells in the connective tissue and an eventual imbalance of osteoblast and osteoclast activity in response to local inflammatory stimuli. In the initial stages of the inflammatory response, innate immune cells such as neutrophils, monocytes/and macrophages respond to the bacterial insult. Subsequently, if the inflammatory condition is not resolved, humoral immune cells, T and B cells, will be activated by antigen-presenting cells (APCs) and will regulate chronic inflammation through constitutive cytokine secretion [34]. Notably, both activated T and B cells in the periodontal tissues are considered to contribute to the disease pathogenesis [43] because these cells can express RANKL and induce osteoclast activity [44]. Indirectly, these cells can also induce the amplification of the pro-inflammatory cytokines (IL-1, IL-6, IL-12, IL-17, IL-18, IL-21, TNF-α, and IFN-γ), which are known to be found in periodontitis [45] and to induce osteoclastogenesis via the RANKL–RANK–OPG pathway. As expected, the accumulation of T and B cells, as well as the levels of the inflammatory cytokines *Il-1β*, *Tnf-α* and *Ifn-γ* (Appendix A), were dramatically elevated in the aged mice compared to the young ones after ligation, and the unligated control group. 

Aging is known to induce significant changes in MSC function, such as by microenvironmental factors (hormonal, immunogenic, metabolic disorder), DNA damage (oxidative stress, telomerase dysfunction), epigenetic alteration (mutation, noncoding RNA), and mitochondrial dysfunction (elevated ROS, mTOR pathway) [46,47,48,49]. As a conventional definition, MSCs have been identified as culture dish adherent spindle shape cells expressing, for instance, CD44, CD90, CD105, CD106, CD166, and Stro-1 and are negative for CD45, CD34, CD14 or CD11b, CD79a or CD19, and HLA-DR surface molecules [50]. Platelet-derived growth factor receptor α (PDGFR-α) [51,52], stem cell antigen 1 (Sca-1), and leptin receptor (Lepr) have also been identified as selective markers of mouse MSCs [51,53]. Our results demonstrated that aged MSCs show lower expression of surface antigens, including Sca-1, CD90, and CD146 (Figure 5C), which was also followed by a lower ability of the aged MSCs to form colonies (CFU-f) compared to young MSCs. These findings are in accordance with previous reports [2].

Several reports have revealed that MSCs could migrate from the perivascular area into the blood circulation in response to signals from damaged tissues, and subsequently, the circulated MSCs may accumulate in the damaged tissues and participate in the regeneration process [50,54,55]. Indeed, our data also showed an increased number of PDGFRα^+^ MSCs in both young and aged mice after ligation. However, the distribution of PDGFRα^+^ MSCs in aged mice was significantly lower than that in the young mice (Figure 4). This could be associated with a higher migration capability of young MSCs compared to the aged MSCs, as demonstrated by the in vitro scratch assay (Figure 5B). These results indicate an impaired functionally of aged MSCs, which could also reflect their impaired local immunotolerance.

Impaired homeostasis of bone metabolism is one of the causes of bone loss in aging. Alteration in MSCs function causes a shift of lineage commitment from osteogenesis to adipogenesis, also leading to a decline in self-renewal capacity [56,57]. Our in vitro data indicated that the expression level of the osteoblast marker genes, *Alp* and *Ocn*, and the ability to form calcium depositions by aged MSCs were significantly reduced compared to young MSCs. Moreover, the expression of adipocyte marker genes, *Pparγ* and *Lpl*, and lipid droplet formation were significantly increased in aged MSCs. These findings indicate the strong commitment of aged MSCs toward the adipogenic lineage, and could be in part related to bone marrow adiposity, an often seen phenomenon in aged mice [58]. On the other hand, the decreased osteogenic capacity of MSCs could be associated with bone-related diseases [59]. Another characteristic of aged MSCs was increased cellular senescence, which is characterized as a state where the cell cycle is irreversibly arrested, negatively affecting the immunomodulatory and differentiation capacities of MSCs [60]. In our experiments, SA-β Gal, which is regarded as the gold standard for evaluating senescence in eukaryotic cells [61], was highly detected in a higher percentage of aged MSCs compared to young MSCs. Notably, a previous study has demonstrated a higher SA-β Gal activity in NK and T cells from older vs. younger human subjects [62]. In this study, however, we were not able to demonstrate the β-Gal activity in the infiltrating B and T cells or in the gingival fibroblasts and epithelial cells surrounding the alveolar bone. Future studies will be necessary to shed more light on the complex mechanism of aging-affected cellular functions, as well as on the interactions of aged cells.

The FAS-L/FAS-mediated cell death pathway is a typical apoptotic signaling in many cell types [63,64,65]. Aging negatively affects the immunomodulatory properties of MSCs. One of the mechanisms of MSC immunoregulation is to induce transient T-cell apoptosis by the FAS-L/FAS pathway [15]. The in vitro experiments evaluating FAS-L expression and MSC co-culture with T-cell importantly revealed a lower expression of FAS-L in MSCs from aged mice (Figure 6C). These results are in agreement with a previous study that also reported the decrease in immunosuppressive capacity in late passage-expanded MSCs, which was correlated with altered membrane glycerophospholipid composition [66]. The decreased immunomodulatory function of MSCs in vitro could be associated with the suppressed immune tolerance and enhanced bone destruction at the periodontitis site.

Clinically, periodontal tissue regeneration therapy is developed via the application of autologous MSCs, growth factors, and/or scaffolds. However, these approaches may have limited applicability because of senescence-induced alterations in the host MSC function, especially in the aged population. This study revealed that understanding the potential molecular mechanisms contributing to a decline in MSC function in aging is crucial when exploring novel strategies to rejuvenate senescent MSCs for the development of novel periodontal regeneration therapies.

In summary, this study investigated the association of aging-affected MSCs and the severity of the periodontitis-associated bone loss in mice. The results demonstrated that aged mice showed more severe periodontal tissue destruction consistent with aging-associated decrease in the expression of stem cell markers and impairment of MSC function, including decreased colony formation, migration, osteogenic differentiation, and, more importantly, reduced immunomodulatory response. 

## 4. Materials and Methods

### 4.1. Animals

C57BL/6J mice at 5 (young) and 50 weeks (aged) of age were purchased from Japan CLEA Co., Ltd. (Tokyo, Japan). The animal experiment protocol used in this study (OKU-2018189) were approved by the Okayama University Research Committee (approved date: 19th April 2018). Throughout the experimental period, all animals were maintained on a standard laboratory diet and in housing under the guideline of the Okayama University Animal Research Committee.

### 4.2. Periodontitis Model

A 5-0 silk thread (Alfresa Pharma, Osaka, Japan) was ligated subgingivally around the mouse mandibular first molar under general anesthesia induced by intraperitoneal injection of 35 mg/kg of 2% xylazine (Celactal, Bayer, Leverkusen, Germany) and 5 mg/kg of ketamine (Ketalar, Daiichi-Sankyo, Tokyo, Japan). The ligatures were maintained in place throughout the experimental period of 10 days. The contra-lateral molar tooth in each mouse was left unligated to serve as baseline control and to decrease inter-animal differences. Mice were sacrificed by cervical dislocation on day-3 and day-10 after ligation. Mandibles were collected and fixed with 4% paraformaldehyde for further analysis.

### 4.3. Micro CT Analysis

The mouse mandible was scanned with micro computerized tomography (micro-CT) (SkyScan1174, Bruker, Kontich, Belgium). The scanning parameters were set to 6.5-pixel size resolution, a peak voltage of 50 kV, and 795 µA with a 0.5 mm aluminum filter. The volumetric reconstruction software NRecon (Bruker) was used to reconstruct the scanned images, which were further mounted in 3-dimensional images by using the volume rendering software, CTVox (Bruker). All images were reoriented and prepared in identical positions for evaluation of alveolar bone loss. The amount of lost alveolar bone was determined by two methods [67]: (i) area: measurement of the area corresponding to the exposed lingual root surfaces of the first mandibular molar as the horizontal limits, and the distance between the cemento–enamel junction (CEJ) and the resorbed alveolar bone crest as the vertical limits (Figure 1D); (ii) depth: measurement of the linear distance from the CEJ to the resorbed alveolar bone crest on the mesiolingual and distolingual regions of the first mandibular molar (Figure 1D). The area and the depth were measured by Image J software (NIH, Bethesda, MD, USA). A single-blinded examiner oriented the images and performed the measurements.

### 4.4. Histological Analysis

After fixation, mandibles were decalcified with 10% ethylenediaminetetraacetic acid (EDTA) and embedded in paraffin, based on a standard protocol. Eight µm-thick sections were prepared and stained with hematoxylin and eosin (HE staining: 1% eosin Y solution, Delafield Hematoxylin: Muto Kagaku Co., Ltd., Tokyo, Japan) or tartrate-resistant acid phosphatase (TRAP) for histological analysis.

### 4.5. Immunohistochemical Analysis

Non-fixed and undecalcified mandibles were freeze-embedded with super cryo-embedding medium (SECTION-LAB Co. Ltd., Hiroshima, Japan) and sectioned in the thickness of 5 µm after mounting the adhesive film onto the sample surface (Kawamoto’s method). The sections were then fixed, blocked with 5% goat serum (Life Technologies, Carlsbad, CA, USA) and stained with rat monoclonal anti-CD45R (B220) antibody (RA3-6B2, eBioscience, San Diego, CA, USA) for B cells, Alexa Fluor 488 rat anti-mouse CD3 (500A2, Biolegend, San Diego, CA, USA) for T cells, goat polyclonal anti-PDGFRα (AF1062, R&D systems, Minneapolis, MN, USA) for MSCs, at 4 °C overnight. After washing, the specimens were incubated with secondary antibody Alexa fluor 488 goat anti-rat IgG (Life Technologies) and Alexa fluor 594 anti-goat IgG (Life Technologies) for 60 min at room temperature. All images were taken by an all-in-one fluorescence microscope (BZ-X700, KEYENCE, Osaka, Japan).

### 4.6. Cell Culture

Primary mouse bone marrow MSCs were isolated from mouse tibia and femur and cultured in basal culture medium with minimum essential medium alpha (α-MEM, Life Technologies) containing 20% fetal bovine serum (FBS, Life Technologies), 2 mM glutamate (Life Technologies), 100 Units/mL penicillin/streptomycin (Sigma-Aldrich, St Louis, MO, USA) and 55 µM 2-mercaptoethanol (2-ME, Life Technologies) in 100 mm dishes (Greiner Bio-One Inc., Frickenhausen, Germany). For isolation of MSCs, the femur and tibia bone marrow were flushed out with 2% FBS/phosphate-buffered saline (PBS) medium through a 70 µm cell strainer (Greiner Bio-One Inc.). The flushed cells were seeded in 100 mm dishes, and after 24 h, the dishes were washed with phosphate-buffered saline (PBS) to eliminate non-adherent cells. After colonization, the attached cells were cultured for 14 days and passaged when they reached a sub-confluent condition. Cells from the second passage were used in the experiments. 

### 4.7. Colony-Forming Unit Fibroblastic (CFU-f) Assay

The flushed bone marrow cells from femurs were seeded into 60 mm dishes (Greiner Bio-One Inc.) at a density of 1 × 10^6^ cells per dish and kept at 37 °C in a 5% CO_2_ humidified incubator. After 14 days, the dishes were rinsed twice with PBS and stained with 0.1% toluidine blue in 1% paraformaldehyde in PBS. More than 50 cells in the colony were counted as a positive colony (CFU-f) using a phase-contrast microscope (BZ-X700, KEYENCE).

### 4.8. In Vitro Wound-Healing Assay

The MSCs from each group were cultured in 6-well plates (Greiner Bio-One Inc.) until confluency. A scratch was made in the middle of the plate with a sterile tip. Cell migration was assessed by measuring the empty space area (%) as the width of the wound gap by using Image J software, based on images taken with a conventional phase-contrast microscope (BZ-X700, KEYENCE).

### 4.9. Cell Surface Antigen Analysis

Flow cytometry was performed for analysis of cell surface antigen expression. The MSCs at passage one were detached using Accutase (Innovative Technologies Inc., San Diego, CA, USA) and resuspend into PBS at a concentration of 1 × 10^5^ cells per 100 µL. The cells were then incubated with fluorescence-labeled primary antibody or isotype control for 30 min in dark condition. The utilized primary antibodies were: FITC anti-mouse Ly-6A/E Sca-1 (Biolegend), PE anti-mouse CD 90, PE anti-mouse CD44, PE anti-mouse CD 146, FITC anti-mouse CD14, PE anti-mouse CD34 (BD, Franklin Lakes, NJ, USA). These samples were analyzed by Accuri™ C 6 Flow Cytometer (BD).

### 4.10. In Vitro Osteogenic and Adipogenic Differentiation

To assess the multiple differentiation potential, MSCs were cultured until sub-confluency and induced to differentiate into the adipogenic or osteogenic lineages for 7 and 21 days, respectively. After that, the gene expression analysis and staining were performed. The adipogenic medium consisted of α-MEM (Life Technologies), 20% FBS (Life Technologies), 2 mM glutamate (Life Technologies), 100 Units/mL penicillin/streptomycin (Sigma-Aldrich) and 55 µM 2-ME (Life Technologies), 10 mM L-ascorbic acid phosphate (FUJIFILM Wako Pure Chemical Corporation, Osaka, Japan), 10 mg/mL insulin (Sigma-Aldrich), 0.5 mM hydrocortisone (Sigma-Aldrich), 6 mM indomethacin (R&D systems), 50 mM isobutylmethylxanthin (Enzo life science, Tokyo, Japan), as reported [66]. Osteogenic medium consisted of α-MEM (Life technologies), 20% FBS (Life Technologies), 2 mM glutamate (Life technologies), 100 Units/mL penicillin/streptomycin (Sigma-Aldrich) and 55 µM 2-ME (Life technologies), 10 mM L-ascorbic acid phosphate (FUJIFILM Wako Pure Chemical Corporation), 200 mM β-glycerophosphate (Sigma-Aldrich), as reported [67]. Mineralization (calcium deposition) and lipid droplet formation were evaluated, respectively, by staining with alizarin red-S or oil red-O, as reported [68,69]. 

### 4.11. Reverse Transcription and Real-Time Reverse Transcription-Polymerase Chain Reaction (RT-PCR)

Gene expression was evaluated by real-time RT-PCR. According to the manufacturer′s instructions, total cellular RNA was extracted from MSCs after being induced to differentiate into the osteogenic and adipogenic lineages by using a Purelink RNA Mini Kit (Life Technologies). MSCs cultured in basal culture medium were used as the control group. 

Total RNA was reverse transcribed using the iScript cDNA Synthesis Kit (Bio-Rad, Hercules, CA, USA). Real-time RT-PCR was performed to quantify the expression of the target gene by using KAPA SYBR FAST qPCR Master Mix (KAPA BIOSYSTEMS, Wilmington, MA, USA) and a CFX96 real-time system (Bio-Rad), as described previously [70]. The expression levels of each mRNA were normalized to that of the reference gene ribosomal protein S29. As the osteogenic specific markers, osteocalcin (*Ocn*) and alkaline phosphatase (*Alp*), and for the adipogenic specific markers, lipoprotein lipase (*Lpl*) and peroxisome proliferator-activated receptor γ (*Pparγ*) were evaluated. Additionally, mRNA levels of inflammation-associated markers (*Il1-β*, *Tnf-α* and *Ifn-γ*) were analyzed in the periodontitis-affected site at day-10 post-ligation. The primer sequences are shown in Table 1.

### 4.12. Senescence Detection Assay 

The MSCs isolated from young and aged mice were seeded in 12-well culture plates (Greiner Bio-One Inc.) and stained with the senescence-associated beta-galactosidase (SA-β Gal) solution using a commercially available kit (Senescence Detection Kit, OZBIOSCIENCES, San Diego, CA, USA). Images were taken with a phase-contrast microscope (BZ-X700, KEYENCE), and SA-β Gal positive cells were counted by Image J software.

### 4.13. Isolation, Activation, and Culture of T Cells

The C57BL/6J mice-derived splenocytes were isolated and cultured with T cell medium, which consisted of Dulbecco′s Modified Eagle Medium (D-MEM: Life Technologies) containing 10% FBS, 2 mM glutamate, 100 Units/mL penicillin/streptomycin, 55 mM 2-ME, 1M 4-(2-hydroxyethyl)-1-piperazineethanesulfonic acid (HEPES: Sigma-Aldrich), 100 mM Sodium Pyruvate (Sigma-Aldrich), 100 x non-essential amino acids (NEAA: Life Technologies). To activate T cells, anti-CD3ε antibody (3 µg/mL, Purified NA/LE Hamster anti-CD3e antibody, BD) pre-coated 60 mm plate (Greiner Bio-one Inc.) and Purified NA/LE Hamster anti-CD28 (2 µg/mL, BD) were used. 

### 4.14. Induction of T Cell Apoptosis

To evaluate the immunomodulatory property of young and aged MSCs, in vitro T cells apoptosis assay was performed. Briefly, MSCs were seeded onto a 12-well plate at a concentration of 2.0 × 10^5^ cells/well and, after 48 h, activated splenocytes (2 × 10^5^ cells/well) were added to the MSCs culture. After 24 h of co-culture, apoptotic T cells were detected by flow cytometry after staining with APC Rat anti-mouse CD4 antibody (1 µg, BD) and a FITC-Annexin V Apoptosis Detection kit I (BD). The expression level of FAS-L in MSCs was evaluated by flow cytometry using the PE Rat anti-mouse FAS-L antibody (BD).

### 4.15. Statistical Analysis

The statistical significance of each data was compared by using the one-way, two-way analysis of variance (ANOVA) and unpaired Student’s *t*-test. Graph Pad Prism 6 (GraphPad Software, La Jolla, CA, USA) was used for the analyses. Significance levels were as follows: * *p* < 0.05, ** *p* < 0.01, *** *p* < 0.001.

## Figures and Tables

**Figure 1 ijms-21-08103-f001:**
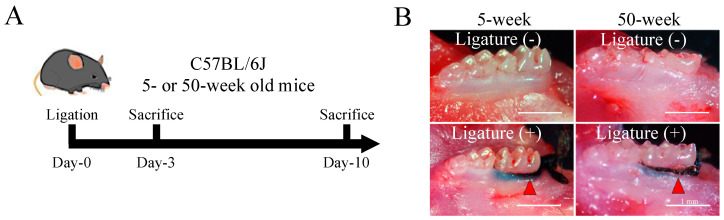
Periodontitis-associated bone resorption is more severe in aged mice. (**A**) Experimental design and timing of sample collection of the ligature-induced periodontitis model in mice. (**B**) Photographs of the control and periodontitis groups showing the ligature in the mandibular first molar of young (5-week-old) and aged (50-week-old) mice. Bar: 1 mm (**C**) Representative micro computerized tomography (micro-CT) images of control and periodontitis groups at day-3 and day-10 after ligation in young and aged mice (red arrow indicated bone loss). Bar: 1 mm (**D**) Bone loss measurement methods. (i) Area: measurement of the area corresponding to the exposed lingual root surfaces of the first mandibular molar as the horizontal limits, and the distances between the cemento–enamel junction (CEJ) and the resorbed alveolar bone crest as the vertical limits; (ii) depth: measurement of the linear distance from the CEJ to the alveolar bone crest on the mesiolingual and distolingual regions of the first mandibular molar, M: mesial, D: distal. Bar: 1 mm and (**E**) results of the micro-CT-based quantitative analysis of bone loss (i) area and (ii) depth, at day-3, and day-10 after ligation in young and aged mice. Note that aged mice showed more severe bone loss compared to young mice, either in the control or ligature-induced periodontitis groups. A two-way ANOVA showed a significant association between age and ligation-induced periodontitis. The bar graph represents the mean ± standard deviation of at least 3 independent samples. * *p* < 0.05, *** *p* < 0.001, two-way ANOVA, Tukey test (*n* = 3).

**Figure 2 ijms-21-08103-f002:**
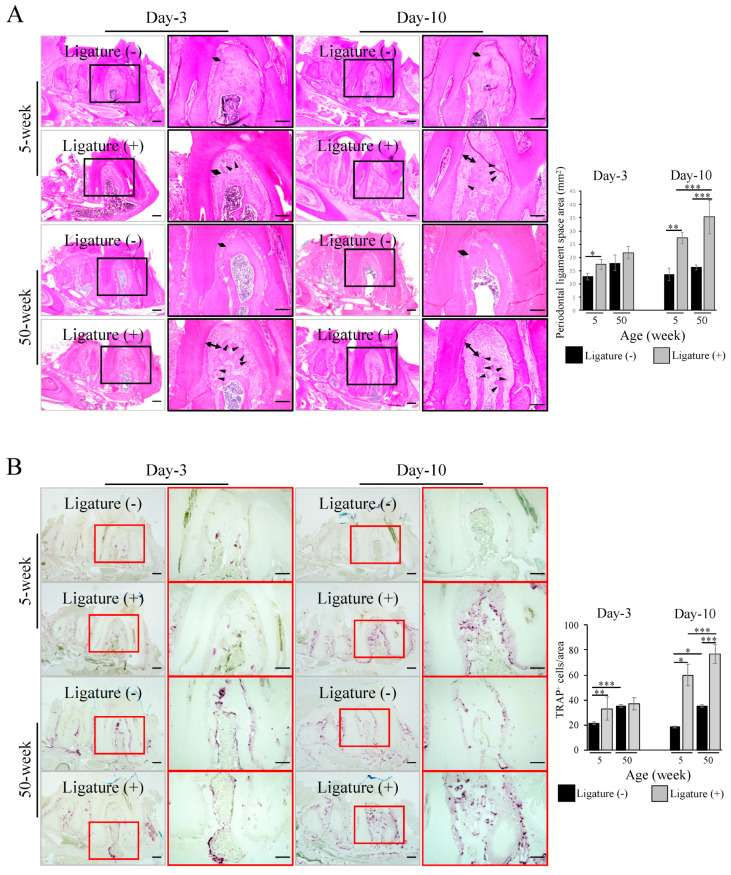
Increased periodontal space and osteoclast activity in the ligation-induced periodontitis in aged mice. (**A**) Hematoxylin and eosin HE staining showing the widening of the periodontal ligament space (bidirectional arrow) in both young and aged mice at day-3 and day-10 after ligation. Note a more severe bone loss and small pits (arrowhead) in aged mice (Black box indicated magnified area in HE staining). Bar: 100 µm. (**B**) Tartrate-resistant acid phosphatase TRAP staining confirming the higher number of TRAP^+^ cells (purple) in the furcation area of the mandibular first molar in aged mice at day-10 after ligation (Red box indicated magnified area in TRAP staining). Bar: 100 µm. The bar graph represents the mean ± standard deviation of at least 3 independent samples. * *p* < 0.05, ** *p* < 0.01, *** *p* < 0.001, two-way ANOVA, Tukey test, *n* = 3.

**Figure 3 ijms-21-08103-f003:**
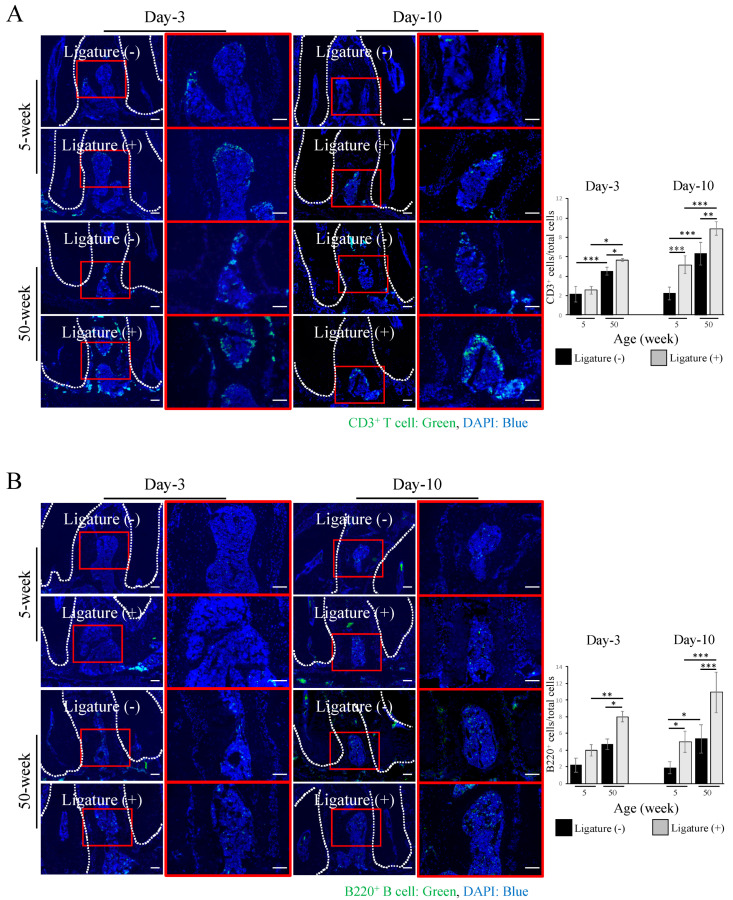
Increased inflammatory cell accumulation at the periodontitis site in aged mice. (**A**) Immunofluorescence images showing that the number of CD3^+^ T cells (green) increased at the furcation area in young and aged mice at day-3 and day-10 after ligation. Cell nuclei were stained with DAPI (blue) Bar: 100 µm. The graph shows the quantitative analysis of cell numbers, indicating a greater number of CD3^+^ T cells in aged mice. (**B**) Immunofluorescence images showing the accumulation of B220^+^ B cells (green) at the furcation area in young and aged mice at day-3 and day-10 after ligation. Cell nuclei were stained with DAPI (blue) Bar: 100 µm. The graph shows the quantitative analysis of cell number, indicating that the number of B220^+^ B cells is significantly higher in aged mice either at day-3 or day-10 after ligation. For (**A**,**B**), the bar graph represents the mean ± standard deviation of at least three independent samples. * *p* < 0.05, ** *p* < 0.01, *** *p* < 0.001, two-way ANOVA, Tukey test, *n* = 3.

**Figure 4 ijms-21-08103-f004:**
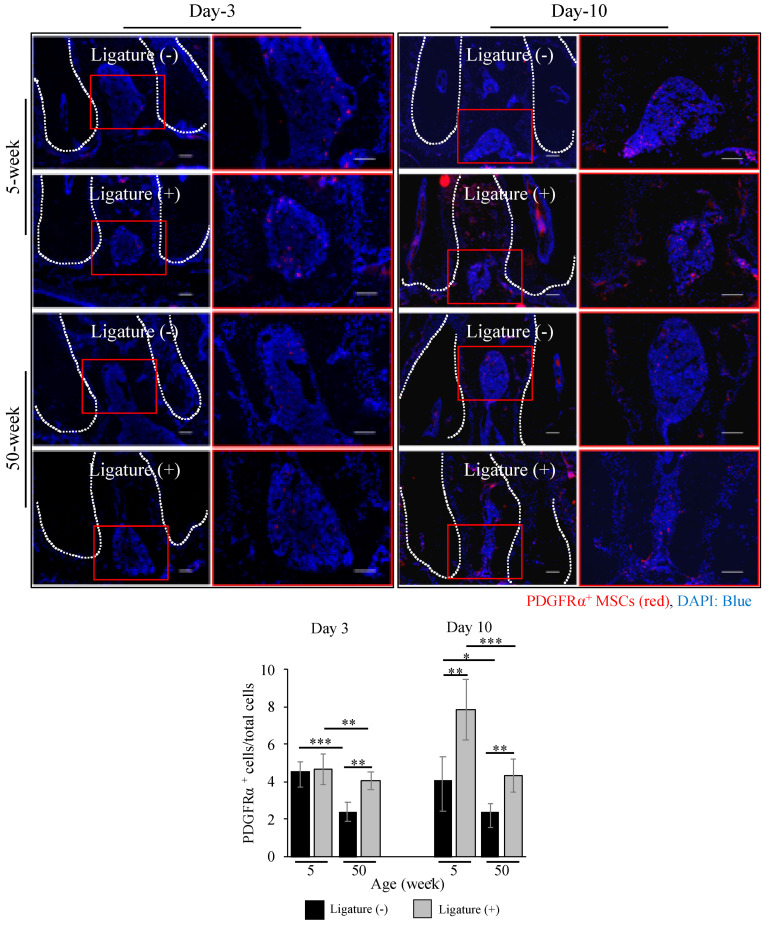
Reduced number of MSCs at the periodontitis site in aged mice. Immunofluorescence images show the number of platelet-derived growth factor receptor α (PDGFRα)^+^ MSCs (red) in the furcation area in young and aged mice. Cell nuclei were stained with DAPI (blue). Bar: 100 µm. The graph shows the quantitative analysis indicating that the number of PDGFRα^+^ Mesenchymal stem cells (MSCs) is decreased in aged mice, more prominently at day-10 after ligation. The bar graph represents the mean ± standard deviation of at least three independent samples. * *p* < 0.05, ** *p* < 0.01, *** *p* < 0.001, two-way ANOVA, Tukey test (*n* = 3).

**Figure 5 ijms-21-08103-f005:**
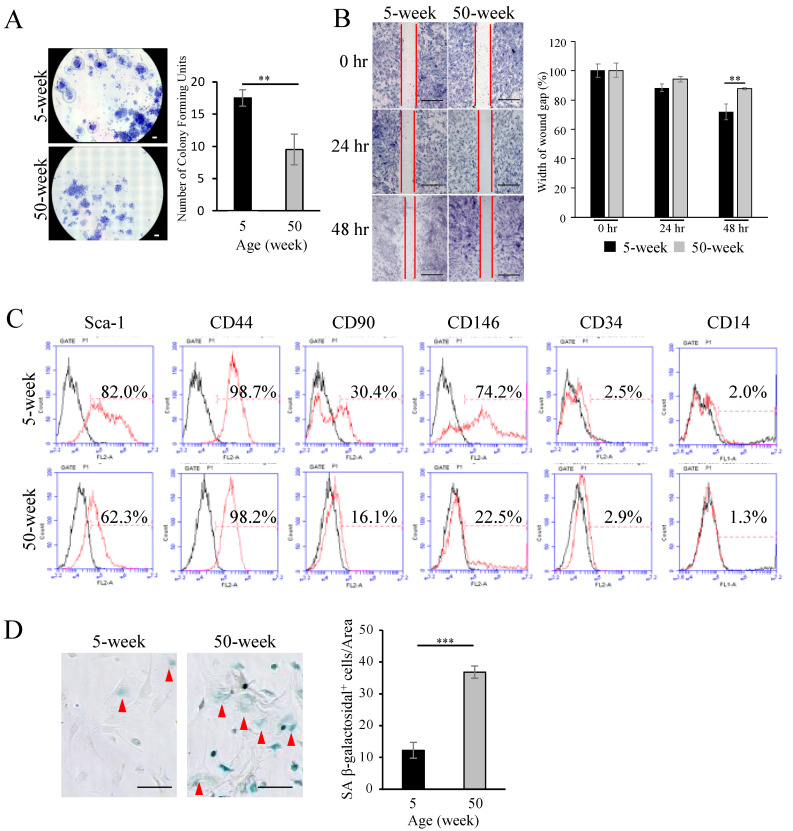
Functional impairment of MSCs from aged mice. (**A**) Photograph and graph showing the quantitative analysis of the number of colonies (CFU-f) formed from flushed bone marrows from young and aged mice. Bar: 500 µm. Note that the CFU-f number is significantly lower in specimens from aged mice. The bar graph represents the mean ± standard deviation of at least 3 independent samples. ** *p* < 0.01, unpaired Student’s *t*-test, *n* = 3. (**B**) Representative images of an in vitro wound healing (scratch) assay using MSCs isolated from young and aged mice. Bar: 500 µm. Note that aged MSCs show slower migration ability compared to MSCs from young mice. The bar graph represents the mean ± standard deviation of at least 3 independent samples. ** *p* < 0.01, two-way ANOVA, Tukey test, *n* = 3. (**C**) Expression of major surface antigen markers for MSCs (Sca-1, CD44, CD90, CD146) in young and aged MSCs determined by flow cytometry. Note that the MSCs show almost no expression of the hematopoietic stem cell markers, CD14 and CD34. Note also that all markers, except for CD44, were significantly decreased in aged MSCs. (**D**) Staining and quantitative analysis of senescence-associated (SA) β-Gal^+^ cells (Red arrow) showing a significantly higher number of senescent cells in aged compared to young MSCs. Bar: 100 µm. The bar graph represents the mean ± standard deviation of at least 3 independent samples. *** *p* < 0.001, unpaired Students’ *t*-test, *n* = 3.

**Figure 6 ijms-21-08103-f006:**
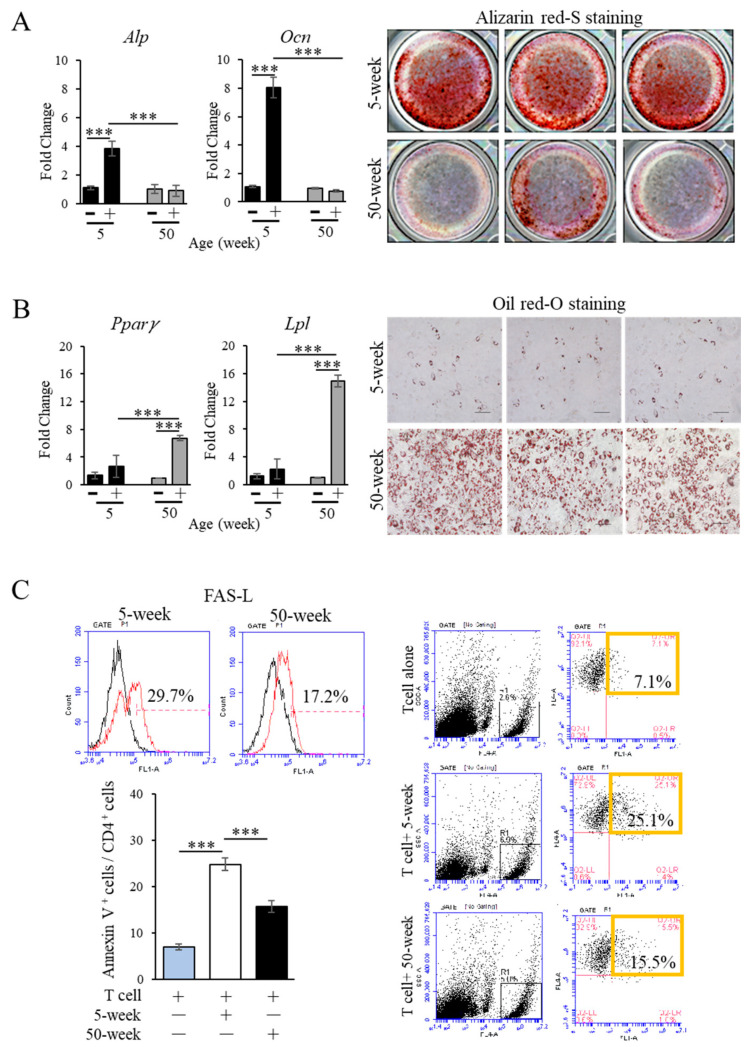
Reduced osteogenic differentiation and immunomodulation in aged MSCs. (**A**) Alizarin red-S staining showing less ability of mineral deposition by aged MSCs compared to young MSCs after osteogenic induction for 21 days. The expression of osteoblast marker genes (*Alp, Ocn*) was determined by real-time RT-PCR. Note the weak staining for alizarin red-S and unchanged expression of *Alp* and *Ocn* in aged MSCs cultured in osteogenic medium for 21 days, indicating the low osteogenic differentiation ability of the aged cells. The bar graph represents the mean ± standard deviation of at least 3 independent samples. *** *p* < 0.001, one-way ANOVA, Tukey test, *n* = 3. *Alp*: Alkaline phosphatase, *Ocn*: Osteocalcin. (**B**) Oil red-O staining showing a dramatically increased formation of lipid droplets by aged MSCs compared to young MSCs. Bar: 100 µm. The expression levels of early adipogenic differentiation markers (*Pparγ* and *Lpl*) were significantly higher in aged than in young MSCs. The bar graph represents the mean ± standard deviation of at least 3 independent samples. *** *p* < 0.001, one-way ANOVA, Tukey test, *n* = 3. *Pparγ*: Peroxisome proliferator-activated receptor *γ*, *Lpl*: lipoprotein lipase. (**C**) Flow cytometric analysis showing a decreased expression of FAS-L in aged MSCs (upper left panel). After co-culture, young and aged MSCs could induce apoptosis of CD4^+^ T cells, as determined by flow cytometric analysis of Annexin V^+^ apoptotic T cells (lower left graph and right panel). Note that the ability of aged MSCs to induce T cell apoptosis is significantly reduced compared to young MSCs. The bar graph represents the mean ± standard deviation of at least 3 independent samples. *** *p* < 0.001, one-way ANOVA, Tukey test, *n* = 3.

**Table 1 ijms-21-08103-t001:** Nucleotide sequence of the primers used for real-time RT-PCR.

Gene	GenBank Registration Number		Primer Set	PCR Product Length (bp)
*s29*	NM_009093	Sense:	5’-GGAGTCACCCACGGAAGTTCGG-3’	108
		Anti-sense:	5’-GGAAGCACTGGCGGCACATG-3’	
*Alp*	JO2980	Sense:	5’-GCTCTCCCTACCGACCCTGTTC-3’	130
		Anti-sense:	5’-TGCTGGAAGTTGCCTGGACCTC-3’	
*Ocn*	BC055868	Sense:	5’-CCAAGCAGGAGGGCAATAAGGTAG-3’	122
		Anti-sense:	5’-CTCGTCACAAGCAGGGTCAAGC-3’	
*Lpl*	BC003305	Sense:	5’-GTCTGGCTGACACTGGACAA-3’	221
		Anti-sense:	5’-TGGGCCATTAGATTCCTCAC-3’	
*Ppar* *γ*	NM_011146	Sense:	5’-GCCAAGGTGCTCCAGAAGATGA-3’	155
		Anti-sense:	5’-CGGGTGGGACTTTCCTGCTAAT-3’	
*Tnf-* *α*	BC117057	Sense:	5’-GTGGAACTGGCAGAAGAG-3’	96
		Anti-sense:	5’-CACAAGCAGGAATGAGAAGA-3’	
*Il1-β*	M15131	Sense:	5’-AATCTCACAGCAGCACATC-3’	194
		Anti-sense:	5’-CCAGCAGGTTATCATCATCAT-3’	
*Ifn-* *γ*	BC119065	Sense:	5’-AGCTCATCCGAGTGGTCC-3’	88
		Anti-sense:	5’-CTCTTCCCCACCCCGAAT-3’	

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
