# Peer review of "Aging-Affected MSC Functions and Severity of Periodontal Tissue Destruction in a Ligature-Induced Mouse Periodontitis Model"

_ijms, 2020, doi:10.3390/ijms21218103_

Round 1

Reviewer 1 Report

In this manuscript entitled “Aging-affected MSC functions and severity of periodontal tissue destruction in a ligature-induced mouse periodontitis model”, Dr Kyaw Thu Aung et al. hypothesized that the decrease in host response and capability of periodontal tissue repair in aged individuals could be associated with aging-induced phenotypic changes in mesenchymal stem cells (MSCs).

Severe bone resorption was induced after ligation in aged mice. This condition was consistent with increased inflammatory cell accumulation at the periodontal bone destruction area, decreased MSCs number with evidence of functional impairment such as decrease in their CFU-f colony forming ability, significantly higher activity of cell senescence-associated β-galactosidase, lower osteogenic differentiation ability and immunomodulatory activity

The results show that aging-induced impairment of MSC function is consistent with progressive periodontal tissue deterioration.

Comments:

The work shows interesting evidences regarding the aging-induced impairment of MSCs in mice.

The orderly presented results are consistent with the hypothesis of a potential correlation of MSCs impairment with progressive periodontal tissue deterioration in a ligature-induced mouse inflammatory model.

However, sentence at manuscript line 332-333:

The results demonstrated that aged mice showed more severe periodontal tissue destruction because of aging-associated decrease in the expression of stem cell markers…

overestimates their significance and should rather read

The results demonstrated that aged mice showed more severe periodontal tissue destruction consistent with aging-associated decrease in the expression of stem cell markers…

Minor issues:

a) Sentence at manuscript line 28:

…significantly lower than young mice.

should read

…significantly lower than in young mice.

b) The first sentence in the Introduction section declares: "Aging is an inherent physiological process of life". Please notice that the eleventh edition of the International Classification of Diseases, released by the WHO in 2018 introduces a new disease code: MG2A Old age. Starting on January 1, 2022, it will be possible to be diagnosed with a condition called "old age".

c) Please reformulate the intricate sentence at manuscript lines 55-59:

Indirect inhibition through the release of anti-inflammatory cytokines, including trransforming growth factor-beta (TGF-β), hepatocytes growth factor (HGF), prostaglandin E2 (PGE2), indoleamine 2,3-dioxygenase (IDO), nitric oxide (NO), or human leukocyte antigen-G5 (HLA-G5) is known to suppress the proliferation and activation of, and inflammatory cytokine secretion by T-lymphocytes, dendritic cells (D.C.s), natural killer cells (N.K.), and B lymphocytes [9-13].

d) Sentence at manuscript line 110:

…in periodontal ligament space in the aged mice (35.28 ± 6.39) than young mice…

should read

…in periodontal ligament space in the aged mice (35.28 ± 6.39) vs. young mice…

Author Response

To the International Journal of Molecular Sciences October, 21st , 2020

Dear Editor-in-chief

We are submitting the revised manuscript entitled: “Aging-affected MSC functions and severity of periodontal tissue destruction in a ligature-induced mouse periodontitis model.” for consideration as an original research article in the International Journal of Molecular Sciences.
Firstly, the authors deeply thank the reviewers for the detailed comments, which significantly helped to improve the quality of the manuscript.
All points addressed by the reviewers have been responded, and the manuscript has been revised according to their comments and suggestions.
Below is the point-by-point response to the reviewer’s comments.

Please see the attachment file.

The authors have no conflicts of interest to declare. This manuscript has not been published or presented elsewhere in part or in entirety and is not under consideration by any other journal.

We are looking forward to receiving a positive reply at your earliest convenience.

Sincerely yours,

Kentaro Akiyama, DDS, PhD.
Senior Assistant Professor
Department of Oral Rehabilitation and Regenerative Medicine, Okayama University Graduate School of Medicine, Dentistry and Pharmaceutical Sciences, Okayama, Japan
2-5-1 Shikata-cho, Okayama, 700-8525, Japan.
Phone: +81 86 235 6682, Fax: +81 86 235 6684
E-mail: akentaro@md.okayama-u.ac.jp

Reviewer 2 Report

This is in response to your request to review the manuscript submitted by Kyaw Thu Aung et al. “Aging-affected MSC functions and severity of periodontal tissue destruction in a ligature-induced mouse periodontitis model”:

The study uses a ligature-induced periodontitis model to compare the role of MSC in the disease process in old (week 50) vs young (week 5 mice).  They present data that shows that the numbers and migratory functions of the MSC are decreased in the aged mice and this is correlated with increased severity of disease (bone resorption) and increased senescence markers on the isolated MSC.  If also true in humans, these results imply that treatment of periodontitis with autologous MSC may not be successful unless the senescent MSC can be rejuvenated.  

Although the paper is generally well written, there are a few things that need attention.

  • Figure 1 E is labeled “Loss of bone area” – this implies that the bone area at the end was subtracted from the bone area at the beginning, but in the legend and the text, they only describe taking one measurement. Perhaps they mean “area with lost bone”?
  • Lines 139-140 are garbled and the meaning is unclear.
  • Figure 5B title says PPAR! – it should be PPAR-g
  • Line 302 “impaired homeostasis in” should be “is”
  • Line 308 awkward wording

In terms of the work itself, the data are strong and the conclusions are supported.  However, the model is an in vivo model, and the authors should acknowledge that the MSC may not be the only cells that have increased senescence in the older animals.  It would be nice to know for example whether the gingival fibroblasts or epithelial cells surrounding the bone or the infiltrating B and T cells also had increased expression of b-galactosidase. 

Author Response

(The authors gave the same response as above.)
